# Visual Perceptual Learning of Form–Motion Integration: Exploring the Involved Mechanisms with Transfer Effects and the Equivalent Noise Approach

**DOI:** 10.3390/brainsci14100997

**Published:** 2024-09-30

**Authors:** Rita Donato, Adriano Contillo, Gianluca Campana, Marco Roccato, Óscar F. Gonçalves, Andrea Pavan

**Affiliations:** 1Department of General Psychology, University of Padova, Via Venezia 8, 35131 Padova, Italy; rita.donato.phd@gmail.com (R.D.); gianluca.campana@unipd.it (G.C.); marco.roccato@studenti.unipd.it (M.R.); 2Elettra-Sincrotrone Trieste S.C.p.A., 34149 Trieste, Italy; adriano.contillo@elettra.eu; 3Human Inspired Technology Research Centre, University of Padova, Via Luzzati 4, 35121 Padova, Italy; 4Brainloop Laboratory, CINTESIS@RISE, CINTESIS.UPT, Universidade Portucalense Infante D. Henrique, 4200-072 Porto, Portugal; oscar@fpce.uc.pt; 5Department of Psychology, University of Bologna, Viale Berti Pichat 5, 40127 Bologna, Italy

**Keywords:** visual perceptual learning, equivalent noise analysis, sampling efficiency, internal noise, glass patterns, modified random-dot kinematograms, non-directional motion

## Abstract

**Background:** Visual perceptual learning plays a crucial role in shaping our understanding of how the human brain integrates visual cues to construct coherent perceptual experiences. The visual system is continually challenged to integrate a multitude of visual cues, including form and motion, to create a unified representation of the surrounding visual scene. This process involves both the processing of local signals and their integration into a coherent global percept. Over the past several decades, researchers have explored the mechanisms underlying this integration, focusing on concepts such as internal noise and sampling efficiency, which pertain to local and global processing, respectively. **Objectives and Methods:** In this study, we investigated the influence of visual perceptual learning on non-directional motion processing using dynamic Glass patterns (GPs) and modified Random-Dot Kinematograms (mRDKs). We also explored the mechanisms of learning transfer to different stimuli and tasks. Specifically, we aimed to assess whether visual perceptual learning based on illusory directional motion, triggered by form and motion cues (dynamic GPs), transfers to stimuli that elicit comparable illusory motion, such as mRDKs. Additionally, we examined whether training on form and motion coherence thresholds improves internal noise filtering and sampling efficiency. **Results:** Our results revealed significant learning effects on the trained task, enhancing the perception of dynamic GPs. Furthermore, there was a substantial learning transfer to the non-trained stimulus (mRDKs) and partial transfer to a different task. The data also showed differences in coherence thresholds between dynamic GPs and mRDKs, with GPs showing lower coherence thresholds than mRDKs. Finally, an interaction between visual stimulus type and session for sampling efficiency revealed that the effect of training session on participants’ performance varied depending on the type of visual stimulus, with dynamic GPs being influenced differently than mRDKs. **Conclusion:** These findings highlight the complexity of perceptual learning and suggest that the transfer of learning effects may be influenced by the specific characteristics of both the training stimuli and tasks, providing valuable insights for future research in visual processing.

## 1. Introduction

Visual perceptual learning is a behavioral method to study long-lasting perceptual improvements resulting from training [1,2,3,4,5]. Practice induces perceptual improvements across different visual domains, such as motion and form perception [3,6], contrast sensitivity [7], and more. The long-term effects of visual perceptual learning are linked to brain plasticity, which refers to the brain’s ability to modify its structure and function [8]. Visual perceptual learning often shows specific improvements tied to trained features such as location and orientation. This specificity has led to contrasting theories in the field: one suggesting localized changes in specific brain regions, and another proposing enhanced interpretation in higher-level areas [9,10]. These opposing views have shaped research for decades. Recent studies have expanded our understanding of these mechanisms. For instance, Ahissar and Hochstein [11,12] proposed a top-down learning process that begins in higher cortical levels and moves downstream to engage the neurons most relevant for stimulus encoding. Simpler tasks are processed at these higher levels, allowing for broader feature transfer, while more complex tasks rely on lower-level cortical areas where neurons are more finely tuned. Furthermore, Jeter et al. [13] found that task precision, rather than difficulty, influences the level of transfer in perceptual learning, suggesting that improvements are more specific to the conditions under which training occurs. The structure and length of training sessions also play a role in generalization, with fewer sessions increasing the likelihood of transfer [14,15]. McGovern et al. [16] contributed to this research topic by demonstrating that perceptual learning can facilitate transfer across different tasks. Their research showed a significant transfer among orientation discrimination, curvature discrimination, and global form tasks, indicating that perceptual learning improvements are not confined to a single task, but can extend across various perceptual domains and tasks. This understanding aligns with the idea that visual perceptual learning involves multiple brain networks [17,18], making it plausible for learning effects to transfer to visual stimuli with similar characteristics. Given this complexity, it is reasonable to suggest that the improvements gained through visual perceptual learning could extend to other stimuli that activate similar neural pathways.

Visual perceptual learning and learning transfer effects have also been used to study the mechanisms underlying form–motion integration and to compare it with directional motion [3]. This is exemplified by our previous study [3], where we conducted an online visual perceptual learning experiment to investigate whether non-directional motion evoked by dynamic Glass patterns (GPs) shares the same processing mechanisms as directional motion evoked by Random-Dot Kinematograms (RDKs). Dynamic GPs are visual stimuli formed by dot pairs called dipoles, and based on their orientation, they can create different global shapes such as translational (i.e., oriented), circular, radial, spiral, etc. [19]. GPs become dynamic when composed of multiple independent frames shown in rapid succession. The frames are independent because dipoles do not follow a specific trajectory throughout the frames (i.e., no dipole-to-dipole correspondence across frames) but maintain a constant orientation, creating an illusory directional motion [3,20,21,22,23,24,25,26,27]. This type of motion has been called *non-directional motion* [3,28]. Conversely, RDKs are composed of single dots that follow a specific trajectory throughout the frames. In our previous study [3], we conducted a ten-session experiment with two distinct groups of participants: one group trained using dynamic GPs, while the other group trained with RDKs. Participants performed a two-interval forced-choice (2IFC) task, where they had to detect which of the two stimuli presented to the screen was the oriented/directional pattern. The results indicated that visual perceptual learning is specific to the trained stimulus, suggesting that directional and non-directional motion may rely on different mechanisms [3]. Despite the growing evidence supporting an interconnected system for motion and form processing, a fundamental question remains: how do these neural networks integrate local form and motion cues to construct a global and coherent perceptual experience? A substantial body of psychophysical research has highlighted the potential of internal noise and sampling efficiency as key parameters in elucidating the interplay between local and global processing of form and motion information [24,29,30,31,32]. Pelli [33] was one of the pioneers of this concept in the field of psychology and psychophysics. Simplifying his model, the author suggested that when we perceive something, our brain adds a consistent amount of internal noise to the sensory input and then processes it to make a decision. By testing individuals’ ability to see against different levels of external noise, we can learn about their sensitivity to that specific visual property tested and how efficiently their brains process that visual information. Within the domain of form and motion perception, internal noise refers to the observer’s ability to detect the inherent variability (variance) in the orientation or motion direction of individual elements within a visual pattern in the absence of any external noise [24,34,35]. Conversely, sampling efficiency reflects the visual system’s capacity to integrate visual information from various spatial and temporal locations into a unified global percept [34]. Both internal noise and sampling efficiency can be estimated using equivalent noise analysis [32,34,35,36,37]. Another prevalent approach in psychophysical research to study global motion (and form) perception has been the use of coherence tasks introduced by Newsome and colleagues [38]. Coherence tasks involve measuring the number of coherently moving or oriented elements (i.e., same direction or orientation) that can be replaced by randomly moving or oriented elements while still allowing for reliable discrimination of the overall direction or orientation, for example, left/right or horizontal/vertical. High motion or orientation coherence thresholds reflect poor global pooling of motion or orientation signals across space [34].

Building on these considerations, while visual perceptual learning has shed light on the mechanisms behind long-lasting perceptual improvements, several critical questions remain unresolved. Our study addresses three key research questions: (i) Does improvement in global form–motion integration transfer to non-directional motion in mRDK? (ii) Does the improvement rely on more efficient noise filtering and/or more efficient sampling? (iii) Are mRDKs easier to discriminate than GPs, even when both stimuli convey non-directional motion? These questions form the foundation of our investigation and are explored in detail in the following sections.

### 1.1. Does Improvement in Global Form–Motion Integration Transfer to Non-Directional Motion from mRDK?

In the current study, we focused on visual perceptual learning in visual local and global processing through two classes of visual stimuli: dynamic translational GPs and a modified version of RDKs that we call modified Random-Dot Kinematograms (mRDKs). To recreate non-directional motion with RDKs, we developed the mRDK so that the individual dots in the coherent portion move either upwards or downwards, while the remaining ones move randomly. Each coherent dot is randomly assigned an upward or downward direction for each frame, resulting in a perceived motion of the dots along the vertical axis (see Appendix A). However, it is not possible to discern an upward or downward trajectory (for further details, please see Section 2.3). Although we attempted to recreate non-directional motion with the mRDK, the main difference between dynamic GPs and the mRDK is that the former is composed of dipoles while the latter is made of single dots. This means that, unlike mRDKs, dynamic GPs involve both motion and form processing elicited by dipole orientation. Visual perceptual learning allowed us to monitor participants’ performance changes on a task involving dynamic GPs to evaluate form–motion global processing. This methodology enabled us to investigate transfer effects onto a structurally different yet perceptually similar visual stimulus (i.e., mRDK). We hypothesized that visual perceptual learning based on a coherence task with dynamic translational GP would cause a gradual decrease in participants’ coherence thresholds, implying that observers enhance their sensitivity to integrate form and motion cues at a global level [3]. We also hypothesized that dynamic translational GPs and mRDKs, being characterized by the same type of non-directional motion, would be processed similarly. Therefore, we expected a learning transfer to the non-trained stimulus.

### 1.2. Does the Improvement of Global Motion–Form Integration Rely upon More Efficient Noise Filtering and/or More Efficient Sampling?

We also aimed to address the following question: what aspects of visual processing are enhanced by learning based on a global coherence discrimination task? Dosher and Lu [39] investigated the mechanisms underlying visual perceptual learning with the aim of determining if distinct mechanisms enhance performance in noisy and clear displays. Two mechanisms—external noise filtering and stimulus amplification—are processes involved in how the human perceptual system adapts to different environments. Their existence has been identified and examined by the authors using a perceptual template model, with specific manipulations of external noise. These mechanisms serve distinct functions: external-noise filtering is crucial in noisy settings, while stimulus amplification is essential in clear environments. Visual perceptual learning associated with these mechanisms reflects improvements in the enhancement of stimulus information quality through external-noise filtering, and/or overcoming intrinsic processing limitations of the human observer through stimulus amplification. The visual stimuli the authors adopted were a simple Gabor and a Gabor with a random noise mask. Participants’ task was to discriminate two different orientations, ±8° from vertical. The results revealed an asymmetric transfer of learning. Training with clear displays improved performance in both clear and noisy environments, suggesting that learning increases the stimulus signal or noise filtering, while training with noisy displays did not benefit performance with clear displays, suggesting that, in this case, learning reduced only the impact of external noise (external noise exclusion) [40]. Using an equivalent noise approach, we tested whether visual perceptual learning could transfer to a task that requires noise filtering and/or sampling efficiency (integration of local signals). According to this approach, all the individual elements (dots or dipoles) can be defined as signals, as they contribute to the global motion/form. This is achieved by assigning the direction/orientation of dots/dipoles based on a Gaussian distribution around a given mean value. The variability in direction/orientation is introduced by varying the standard deviation of the Gaussian distribution [35,41,42,43]. By using an orientation/direction discrimination with variable standard deviation of the Gaussian distribution, we assessed whether visual perceptual learning of dynamic GPs produced changes related to noise filtering at a local level, or sampling efficiency at a global level. It is important to note that in the equivalent noise approach, internal noise is considered even when dealing with external noise, as the standard deviation of direction/orientation affects the internal representation of the stimulus.

### 1.3. Are RDKs Easier to Discriminate than GPs Even When Both Stimuli Convey Non-Directional Motion?

Finally, some studies have shown that RDKs are easier to discriminate than dynamic GPs [3,19,44]. Our final objective was to determine whether this difference is maintained when non-directional motion is introduced into RDKs.

## 2. Materials and Methods

### 2.1. Participants

A total of twelve naïve participants (mean age: 21; SD: 4.427; all females) took part in this experiment. However, two participants were excluded from the final analysis due to their inability to show perceptual learning. All participants had normal or corrected-to-normal vision, and binocular viewing was employed throughout the experiment. This study was conducted in accordance with the tenets of the World Medical Association Declaration of Helsinki [45]. Ethical approval for this study was obtained by the Department of Psychology Ethics Committee at the University of Padova (Protocol number: 4764). Written informed consent was obtained from all participants before the first session.

### 2.2. Apparatus

Visual stimuli were presented on a 23.8-inch Hp Elite E240 monitor (Coimbra, Portugal), with a spatial resolution of 1920 × 1080 pixels and a refresh rate of 60 Hz. Each pixel subtended 1.65 arcminutes. Participants were positioned in a dark room at a viewing distance of 57 cm from the screen. The display of visual stimuli was controlled using MATLAB R2021b Psychtoolbox-3 [46,47,48].

### 2.3. Stimuli

The visual stimuli consisted of dynamic Glass patterns (GPs) and modified Random-Dot Kinematograms (mRDKs). Dynamic GPs consisted of 250 white dipoles, while mRDKs consisted of 500 white dots, each 0.083 deg wide, presented on a gray background. Dipoles and dots were randomly positioned within a circular window with inner and outer radii of 0.2° and 5°, respectively. The distance between the centers of adjacent dots within each dipole was 0.18 degrees [26,49,50]. In the mRDKs, a proportion of dots moved vertically, either upward or downward, while the other dots moved in random directions. The step size for each dot in the mRDKs was 0.18 deg. Specifically, the signal dots were randomly reassigned positions throughout the frames at regular intervals, maintaining a constant vertical directional axis. This produced a flickering texture that created the perception of motion along the vertical axis, without following a specific trajectory. Similarly, in the dynamic GPs, the positions of the signal and noise dipoles (i.e., randomly oriented dipoles) continuously changed throughout the frames, while their orientation remained constant. The frames composing the dynamic GPs and mRDKs were sequentially presented at a rate of 10 Hz, with each frame lasting approximately 0.1 s (see Appendix A).

## 3. Procedure

This study comprised ten sessions, including a 2 h pre-test, eight training sessions of approximately 40 min each [3,51,52], and a 2 h post-test session. Participants completed one training session per day, with a maximum interval of three days between sessions. Both the pre-test and post-test involved four tasks, each consisting of 300 trials: a coherence task with dynamic GPs, a coherence task with mRDKs, an equivalent noise task with dynamic GPs, and an equivalent noise task with mRDKs. The order of tasks was counterbalanced, except for the post-test, which maintained the same task order as the pre-test. The initial coherence of the stimuli was set at 50%.

For the coherence task, participants were presented with two rapid intervals: one containing the coherent translational stimulus (either GPs or mRDKs) and the other containing the noisy stimulus (random dots/dipoles). Participants’ task was to identify which interval contained the coherent stimulus by pressing ‘1’ on the keyboard if they observed the coherent pattern in the first interval and ‘2’ if they observed it in the second interval (2IFC) (see Figure 1a,b). A 1-up/2-down Levitt staircase [53] was used to estimate the 70.7% coherence threshold, calculated as the mean of the last 14 reversals. The steps of the staircase were 25, 20, 15, 10, 5, 2, and 1. These values indicate the extent to which the stimulus coherence is adjusted after each response. If a response is correct, the stimulus coherence is reduced according to these values, progressively adapting to the level of difficulty.

In the equivalent noise task, participants were required to discriminate the perceived orientation or the illusory direction of motion through a two-alternative forced-choice (2AFC) task. Specifically, they were required to indicate whether the moving dots or oriented dipoles were tilted clockwise or counterclockwise from vertical by pressing the left arrow key for counterclockwise perception and the right arrow key for clockwise perception (see Figure 2a,b). The 70.7% discrimination thresholds were assessed using a 1-up/2-down Levitt staircase [53]. However, due to an error in the equivalent noise task staircase, additional computations were necessary, as outlined in the Appendix B.

In coherence tasks, participants can readily discriminate between the signal and noise components. In contrast, within the equivalent noise paradigm, all dots/dipoles are considered as signals, contributing to the overall motion or form. This is achieved by determining the direction or orientation of dots or dipoles based on a Gaussian distribution centered around a specific mean value [35,42,43]. Before starting the pre-test, all participants underwent a familiarization phase with the tasks, performing a series of trials to ensure confidence with the visual stimuli and the experimental procedure. The training sessions focused exclusively on the coherence task with dynamic GPs, comprising two blocks of 300 trials each.

## 4. Results

The analyses and visualizations were conducted using R (v4.4.0; Boston, MA, USA) [54,55].

### 4.1. Coherence Thresholds: Analysis of Pre- and Post-Test

Figure 3 shows the coherence thresholds measured in both pre- and post-test for each type of visual stimulus. The Shapiro–Wilk test revealed that the residuals for coherence thresholds in every condition (i.e., a combination of stimulus and session) deviated from a normal distribution (*W* = 0.756, *p* < 0.001). Each condition’s data also displayed a high positive skewness ≥ 1.7. Additionally, we identified five outliers (80.28, 51.82, 8.77, 89.82, and 80.27) using the Double Median Absolute Deviation (MAD) [56], which were included in the analysis. Due to the non-normality of residuals, we decided to perform a non-parametric factorial ANOVA using the Aligned Rank Transform (ART) method from the R package ARTool [57,58]. ART is a robust non-parametric approach specifically designed for analyzing data when assumptions like normality and equal variances are violated. The ART process involves ranking the data from lowest to highest, assigning each data point a rank based on its relative value within the group. These ranks are then aligned across different conditions by matching ranks of equivalent values, addressing issues related to varying scales or variances among groups. Following alignment, the ranks are transformed back to the original data scale, enabling the application of traditional statistical methods like ANOVA or a linear mixed model. This transformation ensures accurate data analysis while accommodating rank alignment across groups.

Post hoc analysis was conducted using the art.con function [57,59]. Following the rank assignment, we conducted a linear mixed model with session (pre- and post-test) and stimulus (GPs and mRDKs) as within-subject factors, and the intercept across participants functioned as a random effect. Our analysis revealed significant main effects for session (*F*(1, 27) = 4.74; *p* = 0.038) and stimulus (*F*(1, 27) = 8.17, *p* = 0.008), indicating that both factors impact the response variable. However, the interaction between session and stimulus did not reach statistical significance (*F*(1, 27) = 0.25, *p* = 0.621). The analysis revealed that coherence thresholds were lower in the post-test phase than in the pre-test phase, and mRDKs showed significantly higher coherence thresholds compared to GPs.

### 4.2. Equivalent Noise: Analysis of Pre- and Post-Test

For each participant, discrimination thresholds were used to estimate internal noise (σ*_int_*) and sampling efficiency (*η*). This process involves breaking down the total uncertainty in perceiving the stimulus (σ*_obs_*) into two separate components, which are treated independently before being combined quadratically. We followed the approach outlined by Ghin et al. [37], employing the following equivalent noise formula for calculating $\sigma_{obs}$:(1)$$\sigma_{obs}=\sqrt{\frac{{\sigma^{2}}_{int}+{\sigma^{2}}_{ext}}{\eta }}$$

σ*_ext_* refers to the inherent noise in the stimulus, commonly known as external noise, while σ*_int_* represents the intrinsic uncertainty in the observer, referred to as internal noise. The summation is adjusted by a factor *η*, which denotes the effective number of simultaneous samplings conducted by the observer on the stimulus, a process known as sampling. In this study, the equivalent noise parametrization was implemented using a two-point procedure [32]: one with zero external noise (σ*_ext_* fixed at 0) to determine the minimum detectable directional deviation from vertical in the absence of external noise, and another with high external noise to establish the maximum tolerable noise level in terms of orientation/direction deviation from the mean.

The values for zero external noise were obtained from the average of the last 14 reversals from each 1-up/2-down staircase procedure, the uncertainties being the standard deviations of the considered reversals. A different procedure, detailed in the Appendix B, was followed for the high external noise. Here, σ*_obs_* was set at 45° (equivalent to π/4 radians), while σ*_ext_* corresponded to the level of external noise at which an observer achieves 70.7% accuracy in discriminating motion direction. Simplifying the equivalent noise parametrization, at 0 external noise, Equation (1) becomes the following:(2)$$\sigma_{obs,0}=\frac{\sigma_{int}}{\sqrt{\eta }}$$
instead, at high noise, considering $\sigma_{ext,H}\gg \sigma_{int}$, it becomes the following:(3)$$\sigma_{obs,H}≃\frac{\sigma_{ext,H}}{\sqrt{\eta }}$$
combining Equations (2) and (3) can result in the following:(4)$$\eta =\frac{\sigma_{ext,H}^{2}}{\sigma_{obs,H}^{2}}\;\mathrm{and}\;\sigma_{int}=\sigma_{obs,0}\sqrt{\eta }.$$

The uncertainties associated with η and $\sigma_{int}$ can be computed as follows:(5)$$\delta \eta =\frac{2\sigma_{ext,H}}{\sigma_{obs,H}^{2}}\delta \sigma_{ext,H}\;\mathrm{and}\;\delta \sigma_{int}=\sqrt{\eta {\left(\delta \sigma_{obs,0}\right)}^{2}+\frac{\sigma_{obs,0}}{4\eta }\left(\delta \eta^{2}\right)}$$

Based on the extracted values, we conducted an analysis to compare the sampling efficiency and internal noise for GPs and mRDKs to observe the impact of learning transfer. Figure 4 shows the results of sampling efficiency measured in both pre- and post-test. The Shapiro–Wilk test indicated that the residuals deviated from a normal distribution only for one condition (*W* = 0.66, *p* = 0.0003), displaying a high positive skewness in the same condition 2.17 (SE = 0.69) and a kurtosis of 3.74 (SE = 1.33). We also detected eight outliers (3.84, 7.58, 5.15, 3.86, 3.90, 7.45, 8.19, and 5.07). Due to the presence of outliers and non-normally distributed residuals in one condition, we employed again the ARTool package. Following rank assignment, we performed a linear mixed model with session (pre- and post-test) and stimulus (GPs vs. mRDKs) as within-subjects factors. The random effect was always the intercept across participants. The analysis revealed that the main effect of the session is not significant (*F*(1, 27) = 3.48, *p* = 0.07), as well as the main effect of stimulus (*F*(1, 27) = 0.01, *p* = 0.91). However, a significant interaction between session and stimulus was observed (*F*(1, 27) = 4.85, *p* = 0.03). Subsequent Holm-corrected post hoc comparisons were conducted to further investigate the interaction effect (Holm correction applied for six tests). A significant difference was identified in the comparison between GPs and mRDKs in the post-test session (*p_adj_* = 0.027), as well as between pre- and post-test for mRDK (*p_adj_* = 0.029). These results highlight the substantial impact of visual perceptual learning on sampling efficiency, revealing that sampling efficiency in fact improved only for the non-trained mRDK stimulus. Specifically, for GPs, there was no significant difference in terms of simultaneous samplings conducted by the observer on the stimulus (pre-test: M = 2.26, SEM = 0.14; post-test: M = 2.62, SEM = 0.21). In contrast, for the mRDKs, a significant reduction in simultaneous samplings was observed (pre-test: M = 3.38, SEM = 0.27; post-test: M = 1.1, SEM = 0.07).

Figure 5 shows the variation in internal noise between pre- and post-tests for both GPs and mRDKs. The Shapiro–Wilk test revealed that the residuals were not normally distributed in three out of four conditions (all *p* < 0.01), showing a positive skewness of 2.35. Four outliers were identified with values of 2.52, 2.87, 1.69, and 2.35. To analyze the internal noise values, the ARTool method was employed. After rank assignment, a linear mixed model was performed, revealing significant effects for the session (*F*(1, 27) = 4.24, *p* = 0.049) and stimulus (*F*(1, 27) = 5.168, *p* = 0.031). However, the interaction effect between session and stimulus was not statistically significant (*F*(1, 27) = 1.535, *p* = 0.225). Examining the impact of visual perceptual learning, a significant decrease in internal noise was observed from the pre-test to the post-test (pre-test: mean = 0.62, SEM = 0.044; post-test: mean = 0.29, SEM = 0.02). Furthermore, higher levels of internal noise were found for mRDK compared to GPs (mRDK: mean = 0.7, SEM = 0.045; GP: mean = 0.22, SEM = 0.01).

### 4.3. Learning Curves

Figure 6 depicts the average coherence thresholds estimated across the eight learning sessions with GPs. The starting and ending points on the curve represent the contrast thresholds estimated before and after training. To analyze the learning curve and investigate the relationship between learning sessions and coherence threshold, three different models were examined: linear, power, and exponential. The models’ forms are described as follows.

These functions were fitted to the learning curve, and the best-fitting model was determined to be the power function based on the lowest values of AIC, AICc, and BIC (refer to Table 1). In the power function, parameter ‘a’ represents the scale parameter, indicating the power function’s value at x = 1 (initial assessment), while ‘b’ represents the learning rate, with smaller values suggesting slower progress across sessions; ‘x’ denotes the extent of practice (learning sessions). In this study, the parameters were estimated as *a* = 33.89 and *b* = 0.20. The power fitting model indicates that the rate of improvement in learning performance does not follow a linear pattern with the number of practice sessions. This nonlinearity suggests that the learning rate is higher at the beginning of the training and decreases as the number of sessions increases. The linear and exponential models did not fit as well because the linear model assumes a constant rate of improvement over time, and the exponential model assumes rapid early gains that continue exponentially. Neither model captured the gradual slowing of the learning rate observed in our data. The power function, with its flexibility, better modeled this nonlinear learning process.

## 5. Discussion

The primary objective of this study was to further investigate the impact of visual perceptual learning on visual stimuli containing form and motion cues that evoke non-directional motion. We aimed to examine the learning transfer effects to a different, yet similar, stimulus, as well as to a different task. To achieve this, we estimated coherence thresholds for dynamic Glass patterns (GPs) during a motion discrimination task across ten sessions, where participants identified in which of the two presented intervals the coherent pattern appeared. Additionally, to assess learning transfer at both the stimulus and task levels, we introduced tasks in the pre- and post-test to evaluate coherence thresholds for modified Random-Dot Kinematograms (mRDKs) and discrimination thresholds for both high- and zero-noise levels, computing internal noise and sampling efficiency for both dynamic GPs and mRDKs. This study is the first to test a new class of visual stimuli that simulates the illusory directional motion elicited by mRDKs in the absence of form cues. This approach was crucial for better understanding the dynamics underlying learning transfer to a different visual stimulus.

As reported in the introduction, our previous visual perceptual learning study [3] showed that eight days of training on dynamic translational GPs did not lead to learning transfer to directional RDKs, suggesting that distinct mechanisms may underly directional and non-directional motion processing. In contrast to directional RDKs, where each dot follows a specific trajectory, the current study found not only significant learning effects on the trained stimulus (i.e., dynamic GPs) and task (i.e., coherence thresholds) but also substantial learning transfer to the non-trained stimulus (i.e., mRDKs). This suggests that the processing mechanisms underlying two forms of non-directional motion—one induced by dynamic GPs, which integrate both form and motion cues, and the other by mRDKs, which contain only ambiguous motion cues—share overlapping processing mechanisms.

We also observed partial learning transfer to a different task, evidenced by improved internal noise filtering after eight days of training on coherence thresholds. According to signal detection theory, internal responses to stimuli are probabilistic, meaning that a particular stimulus has only a certain probability of triggering a specific internal response [60,61]. In the context of perceptual learning, each trial elicits an internal response in the observer that may be based on decreasing the internal noise, enhancing processing efficiency (external noise exclusion), or both. Our findings indicate that training on global features discrimination tasks can enhance local noise filtering. However, the ability to integrate local information into a global percept did not achieve statistical significance (*p* = 0.07). Nonetheless, the trend observed suggests that a larger sample size may help clarify this effect. Our results also show that the variance in sampling efficiency and internal noise decreases drastically between pre- and post-training sessions. This may reflect an optimization of perceptual processes, such as local and global cues processing, leading to more accurate and reliable performance on visual discrimination tasks.

A slightly different outcome was observed in a study by Gold et al. [60], where the authors investigated visual perceptual learning using the signal detection theory. Their aim was to determine whether perceptual improvements resulted from increased internal signal strength or decreased internal noise. They employed an external noise masking and a double-pass response consistency method (which measures how reliably participants provide the same answers when repeating the same task under identical conditions) to analyze how observers learned to detect unfamiliar visual stimuli. Specifically, the tasks involved discriminating between two types of unfamiliar patterns: human faces and abstract textures. Although participants’ performance improved with practice, internal noise filtering did not change. This suggested that learning enhanced internal signal strength without decreasing internal noise. Similarly, Kurki and Eckstein [62] used a classification image methodology to investigate which parts and features of the stimulus the visual system processes at different stages of learning. They found that while sampling efficiency increased with training, internal noise filtering was not affected. These differences between studies may arise from the distinct methodologies and models used, suggesting that future research should explore the variations between these models in the context of visual perceptual learning.

Furthermore, we observed a significant interaction between the visual stimulus and the session, as well as a generalization of learning to the non-trained stimulus (mRDK) for sampling efficiency. This indicates that the impact of the training session varied depending on the type of visual stimulus, revealing distinct effects in the post-test tasks. In other words, for sampling efficiency, the training had a different impact on participants’ performance depending on whether the stimulus was a dynamic GP or an mRDK. It is important to consider that the equivalent noise model is a mathematical framework, and we did not specifically train participants on any equivalent noise tasks. Since sampling efficiency for dynamic GPs was already at floor level in the pre-test, there was likely little room for improvement, suggesting that the task with dynamic GPs was relatively simple. For the mRDK, being a more challenging stimulus to discriminate, there might have been a small generalization effect. These different effects suggest that the type of visual stimulus plays a critical role in how training influences visual perceptual learning and the ability to integrate the orientations/directions of dynamic GPs and mRDKs.

Finally, we found a statistical difference between dynamic GPs and mRDKs, with mRDKs showing higher coherence thresholds compared with dynamic GPs. This indicates that participants could more easily discriminate the motion axis orientation in dynamic GPs than in mRDKs. This could be attributed to the increased difficulty in processing mRDKs due to the random scrambling of the positions of individual elements within the pattern, as mRDKs introduce visual noise and disrupt a coherent motion direction, thus requiring more complex computation to be perceived. Therefore, we can conclude that the well-defined directional motion in directional RDKs makes them easier to process than dynamic GPs, as supported by previous investigations [3,19,44].

## 6. Conclusions

This study examined the impact of visual perceptual learning on stimuli that integrate both form and motion cues, with a particular focus on non-directional motion. The primary aim was to evaluate the transferability of learning across different stimuli and tasks. The results showed that training on dynamic Glass patterns (GPs) not only improved performance on the trained task (coherence threshold) but also transferred to an untrained stimulus—modified Random-Dot Kinematograms (mRDKs). This suggests that both types of non-directional motion share common processing mechanisms. Additionally, partial transfer to a different task was observed, as reflected by enhanced internal noise filtering.

Participants also found dynamic GPs easier to process than mRDKs.

Overall, these findings deepen our understanding of visual perceptual learning and its transfer mechanisms in non-directional motion perception, highlighting the importance of stimulus characteristics and the potential for visual perceptual learning to enhance internal processing efficiency.

## Figures and Tables

**Figure 1 brainsci-14-00997-f001:**
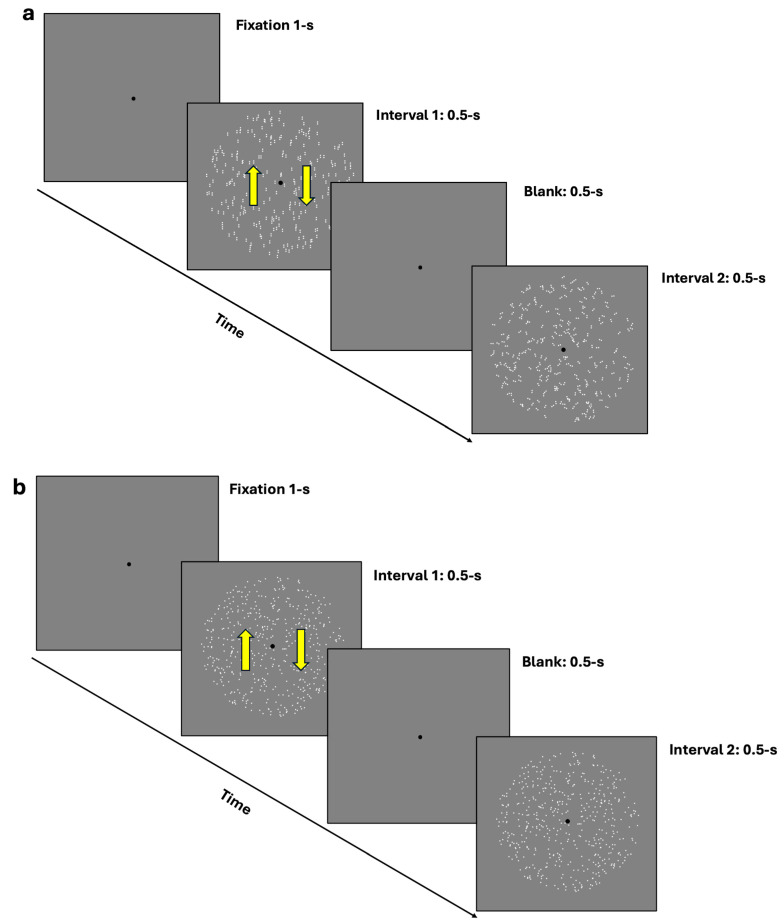
(**a**,**b**) depict the 2IFC task procedure. (**a**) illustrates the procedure using dynamic GPs, where the first interval contains a vertically oriented GP, and the second interval contains a random/noisy GP. (**b**) depicts the procedure using mRDKs, with bidirectional movement along the vertical axis. In both figures, the arrows in the first interval indicate the bidirectional illusory motion along the vertical axis. The interval order shown in the figure is just an example; in the actual experiment, the coherent stimulus could randomly appear in either the first or the second temporal interval. Additionally, for illustrative purposes, the stimuli are shown at the maximum level (100%) of their coherence in this figure.

**Figure 2 brainsci-14-00997-f002:**
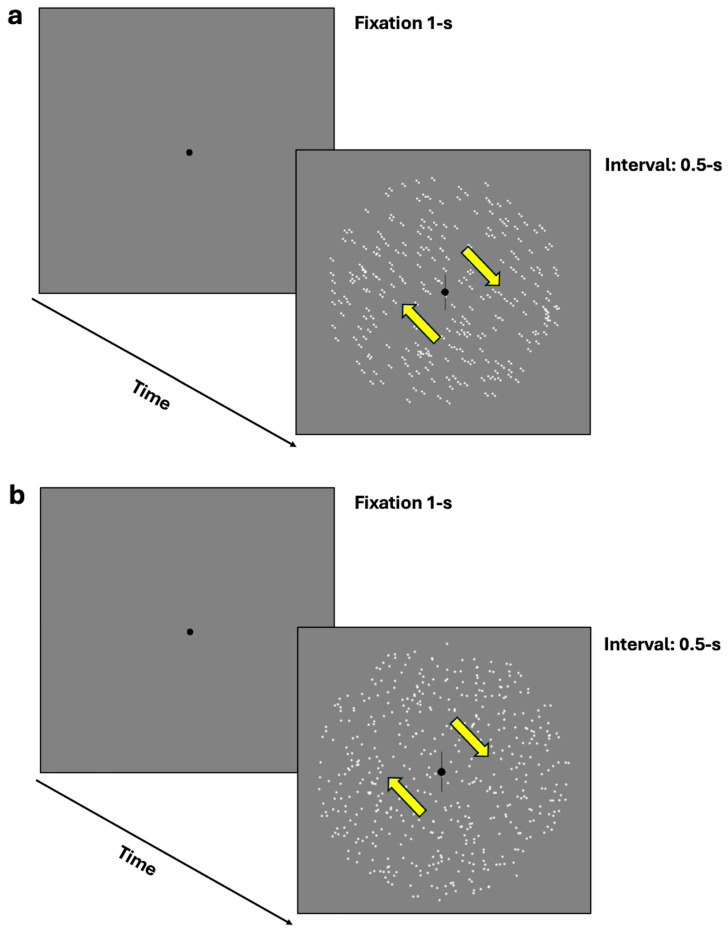
(**a**,**b**) represent the equivalent noise task where participants were required to discriminate the perceived orientation or illusory direction of motion (either clockwise or counterclockwise from vertical) using a two-alternative forced-choice task (2AFC). (**a**) depicts the procedure with dynamic GPs, while (**b**) illustrates the mRDKs. The arrows represent the bidirectional illusory motion along the oblique axis and were not displayed during the experiment.

**Figure 3 brainsci-14-00997-f003:**
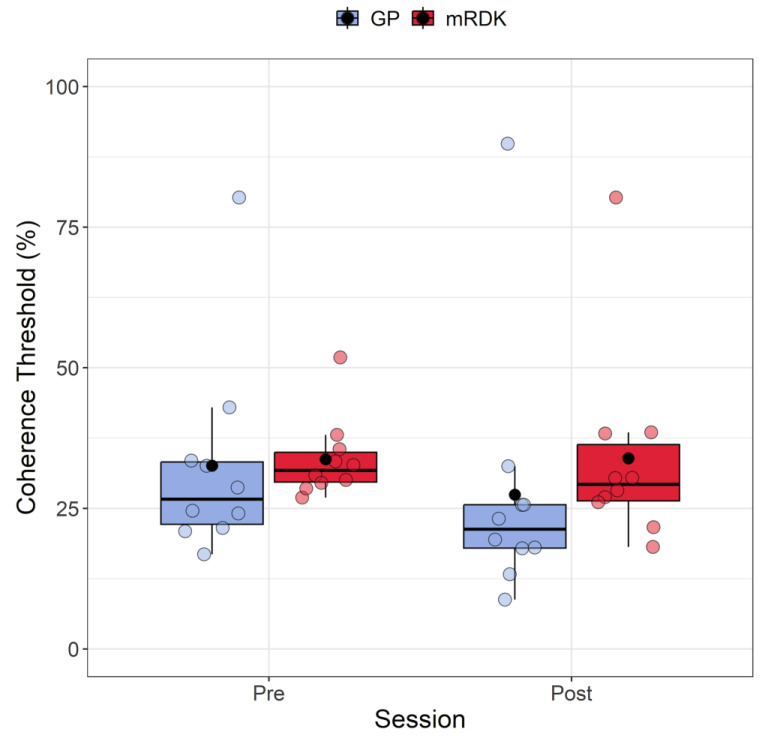
Boxplots depicting coherence thresholds for pre- and post-test sessions, as well as for GPs and mRDKs. Each box in the plot represents the interquartile range (IQR) of the data, with the median indicated by the horizontal line inside the box. The whiskers extend to the minimum and maximum values within 1.5 times the IQR from the first and third quartiles, respectively. Additionally, the black point inside each box denotes the mean of that condition. The colored dots represent individual data points, with blue indicating dynamic GPs and red indicating mRDK.

**Figure 4 brainsci-14-00997-f004:**
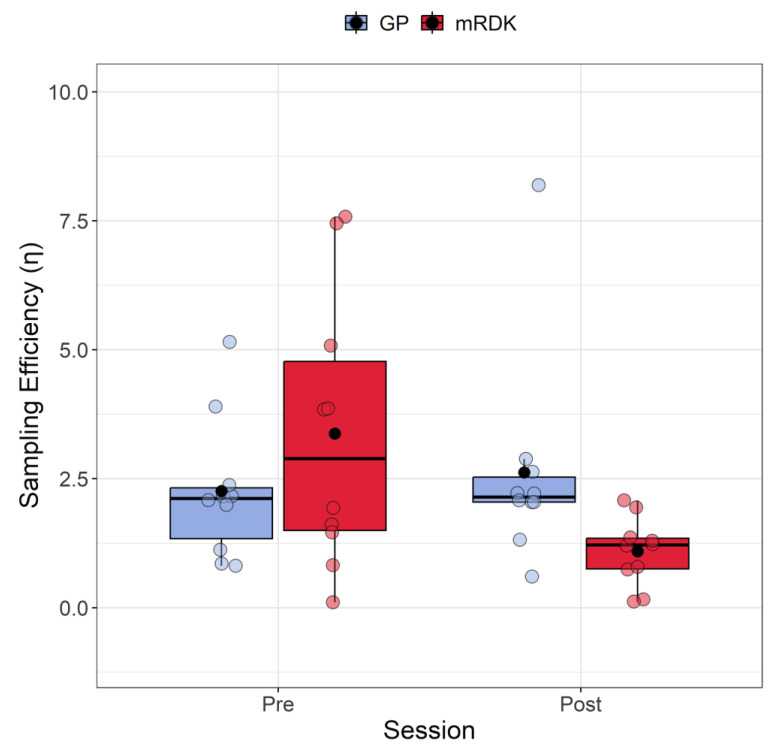
Boxplots of sampling efficiency (η) for pre- and post-test conditions. For each boxplot, the horizontal black line indicates the median, whereas the dot within each box represents the mean sampling efficiency for each condition. The colored dots represent individual data points, with blue indicating dynamic GPs and red indicating mRDK.

**Figure 5 brainsci-14-00997-f005:**
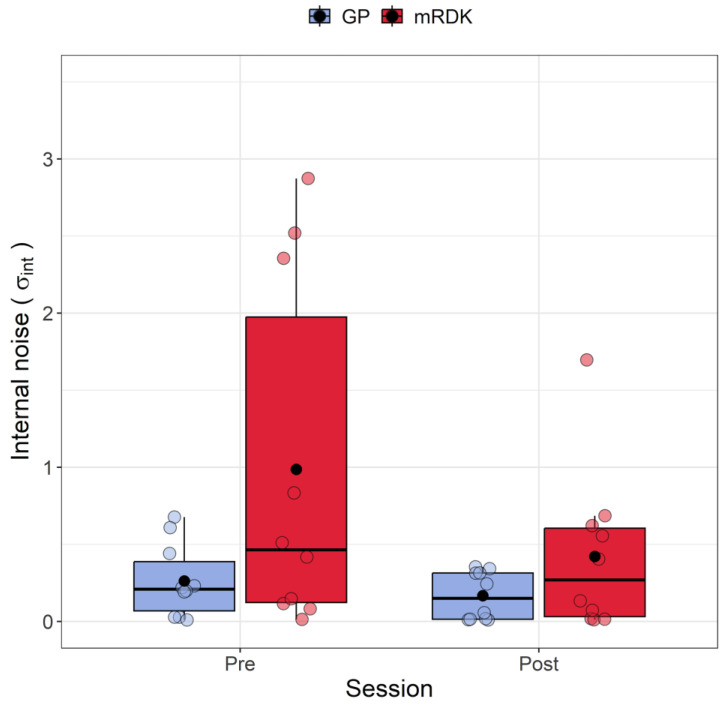
Boxplots of internal noise ($\sigma_{int}$) for pre- and post-test conditions. For each boxplot, the horizontal black line indicates the median, whereas the dot within each box represents the mean internal noise (in radians) for each condition. The colored dots represent individual data points, with blue indicating dynamic GPs and red indicating mRDK.

**Figure 6 brainsci-14-00997-f006:**
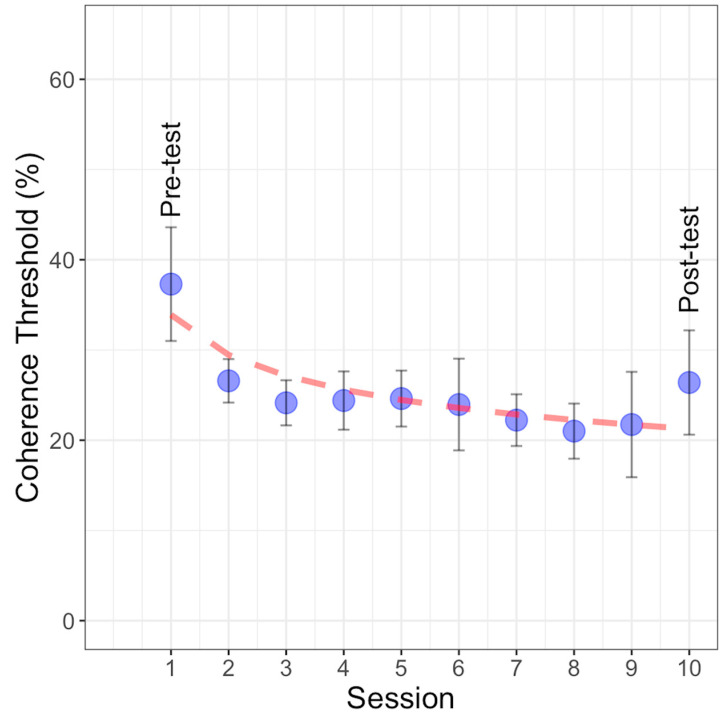
Coherence threshold percentages with GP stimuli observed across ten experimental sessions, including the pre-test, post-test, and eight training sessions. The blue points represent the mean coherence threshold for each session, with error bars indicating ±1SEM. The dashed red line represents the power function fit to the data.

**Table 1 brainsci-14-00997-t001:** Functions used to fit the coherence threshold values and associated estimators of prediction error.

Model Name	Model Function	Estimators of Prediction Error
Linear function	y=ax+b	AIC = 59.12 AICc = 63.15 BIC = 60.06
Power function	$y={ax}^{-b}$	AIC = 52.08 AICc = 56.08 BIC = 52.98
Exponential function	$y={ae}^{bx}$	AIC = 58.5 AICc = 62.5 BIC = 59.41

## Data Availability

The data presented in this study as well as the MATLAB scripts used for the experiment are openly available in Open Science Framework (OSF) at https://osf.io/whtpd/?view_only=4229c09c807d47beb88dbab6b78a7681 (Accessed on 27 September 2024).

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
