# Peer review of "Visual Perceptual Learning of Form–Motion Integration: Exploring the Involved Mechanisms with Transfer Effects and the Equivalent Noise Approach"

_brainsci, 2024, doi:10.3390/brainsci14100997_

Round 1

Reviewer 1 Report

Comments and Suggestions for Authors

This paper “Visual perceptual learning of form-motion integration: exploring the involved mechanisms with transfer effects and the equivalent noise approach”.

The topic is justified. The paper could be further improved if the following remarks are taken into consideration:

1.       The title of the study seems appropriate to the conducted research activity.

2.       ABSTRACT: overall abstract is well organized.

3.       Few grammatical mistakes were found in the whole draft of the article; the authors need to fix these.

4.       Introduction section few of the references are outdated.

5.       Add a recently conducted related to establishing the research problem.

6.       Materials and methods, twelve naïve participants seem less in volume to establish generic findings, while two were excluded, hence only 10 were used in the research.

7.       Results show that the best-fitting model was determined to be the power function based on the lowest values of AIC, why other two functions (linear and exponentials) were not fitted best?

8.       The motivation of the study is not explicitly mentioned.

Comments on the Quality of English Language

minor edits required

Reviewer 2 Report

Comments and Suggestions for Authors

Summary

In the manuscript “Visual perceptual learning of form-motion integration: exploring the involved mechanisms with transfer effects and the equivalent noise approach” the authors investigate how visual perceptual learning (VPL) for on class of motion stimuli (glass patterns) transfers to modified random dot kinematograms (mRDKs). Previous work by the authors has failed to find GP transfer to RDKs; however, this could be because traditional RDKs contain directional motion. In contrast mRDKs are more closely aligned with GPs as they can be considered to contain illusory, non-directional motion. Using a standard perceptual learning paradigm (relatively few participants, several sessions of training), the authors find improvements in the trained stimuli, as predicted. Importantly, the authors also find transfer to mRDKs, providing some insight into the mechanisms underlying VPL of motion and form. Finally, the authors found an interaction between stimulus type and sampling efficiency, with GPs being affected differently than mRDKs.

Evaluation

I enjoyed reviewing this manuscript. The findings are presented clearly, the question of transfer in the context of perceptual learning is important (and contentious, as the authors note), and I think the findings make a nice contribution to this literature. I have a few minor suggestions for improvement, listed below:

1. General: I would encourage the authors to review their manuscript closely to remove unnecessary acronyms. Frequent terms like GP and mRDK seem reasonable, but there are many other acronyms (VPL, EN) that could likely be removed and improve the readability of the paper.

2. It felt like the abstract ended abruptly. I would encourage the authors to add a final sentence commenting on the implications of their findings.

2. P2 L55 “Given that VPL involves the activation of brain networks” – I am not sure what this clause really adds, as presumably any perceptual process or behavior would involve the activation of brain networks. Perhaps the authors mean multiple brain networks?

3. P3 L75 Please consider revising the opening to this paragraph to make it clear from the beginning that you are describing previous research and not the present study. At first, it is unclear that you are describing a previous study, and I was quite confused at first when you then stated that no transfer was found.

4. P15 L465-467 I suggest that the authors find a different way to discuss their marginal results. Significance in null hypothesis significance testing is a binary state, and thus it feels odd to describe a “hint” of significance.
